# Effects of Oral Health Interventions in People with Oropharyngeal Dysphagia: A Systematic Review

**DOI:** 10.3390/jcm11123521

**Published:** 2022-06-19

**Authors:** Lianne Remijn, Fabiola Sanchez, Bas J. Heijnen, Catriona Windsor, Renée Speyer

**Affiliations:** 1Academy of Health, HAN University of Applied Sciences, 6525 EN Nijmegen, The Netherlands; 2Pedagogical Professional Team District Frogner, 0201 Oslo, Norway; fabiolisanchez@gmail.com; 3Department Special Needs Education, Faculty of Educational Sciences, University of Oslo, 0318 Oslo, Norway; catrionawindsor@hotmail.com (C.W.); renee.speyer@isp.uio.no (R.S.); 4Department Otorhinolaryngology and Head and Neck Surgery, Leiden University Medical Center, 2300 RC Leiden, The Netherlands; b.j.heijnen@lumc.nl; 5Curtin School of Allied Health, Faculty of Health Sciences, Curtin University, Perth, WA 6102, Australia

**Keywords:** oral hygiene, deglutition swallowing problems, aspiration pneumonia, efficacy, treatment, PRISMA

## Abstract

People with oropharyngeal dysphagia (OD) are at risk of developing aspiration pneumonia. However, there is no “best practice” for oral health interventions to improve swallowing-related outcomes, the incidence of aspiration pneumonia, and oral health in people with OD. Systematic literature searches were conducted for oral health interventions in OD in PubMed, Embase, CINAHL, and PsycINFO until July 2021. Original articles published in English and reporting pre- and post-intervention measurements were included. The methodology and reporting were guided by the PRISMA checklist. The methodological quality of the eight included studies was rated using the QualSyst critical appraisal tool. The oral health interventions in people with OD were diverse. This study shows little evidence that regular oral care and the free water protocol or oral disinfection reduced the incidence of aspiration pneumonia in people with OD. Oral cleaning, twice a day with an antibacterial toothpaste in combination with intraoral cleaning or the free water protocol, proved to be the most promising intervention to improve oral health. The effect of improved oral health status on swallowing-related outcomes could not be established. Increasing awareness of the importance of oral health and implementing practical oral care guidelines for people involved in the daily care of people with OD are recommended.

## 1. Introduction

Oropharyngeal dysphagia (OD) may occur in people of all ages and is estimated to affect about 8% of the world population (590 million people) [1]. The prevalence estimates of OD in elderly populations are over 30% in patients after stroke and 60 to 80% in patients with neurodegenerative diseases [2]. Reports of OD prevalence in children with multiple neurodevelopmental disabilities range from 85 to 89% [3,4] and may be as high as 99% in children with severe cerebral palsy [5]. OD increases the risk of malnutrition and dehydration due to insufficient oral intake and is frequently associated with severe distress during meals, aspiration, and aspiration pneumonia [6]. Moreover, OD contributes to a decreased functional health status and quality of life and an increased risk of mortality [6]. One systematic review of patients with OD found their durations of hospital stays increased by 2–8 days and their health care costs were approximately 40% higher than in patients without OD [7]. 

Aspiration pneumonia, one of the most critical complications of OD [8,9], is an infectious process caused by the aspiration of oropharyngeal secretions composed of saliva and food residues and containing oral pathogens [10]. Due to poor oral hygiene, saliva contaminated with an increased quantity of multiple bacteria species can harbor microbes that if colonized and aspirated may result in bacterial pneumonia [9,11,12,13]. Saliva contains both nutrients for microbial growth and for anti-microbial components preventing the disruption of the oral microbiome [14]. The decreased salivary clearance in OD may result in the growth of oral pathogens, contributing to reduced oral health [11,15,16]. Moreover, more frequent residuals of microbes remain in the lower airways due to the impaired cough reflex of OD [10]. Consequently, the combined impacts of poor oral hygiene and OD may increase the risk of aspiration pneumonia.

Oral health can be defined as being free from mouth and facial pain, and diseases and disorders that affect oral cavity functioning and is considered a key indicator of overall health, well-being, and quality of life [17]. There is evidence that supports the use of oral health care interventions in preventing oral diseases [15], reducing the frequency of pneumonia [14,18], and improving quality of life [13]. Interventions in oral health are typically provided by oral health care professionals and include a combination of instructions for effective tooth brushing or denture cleaning techniques, the use of a mouth rinse, and ensuring that regular dental check-ups are maintained [19]. Meanwhile, in nursing homes, usual oral care is generally less intensive and self-administered or provided by people who have not received specialized oral hygiene training [20]. In populations with increased rates of OD (e.g., cerebral palsy [3], dementia, stroke, or people over 85 years [21]), the activities of oral and dental care may be challenging, especially in populations who require support in carrying out activities of daily living [11,22]. As oral health is often considered a low priority relative to other competing demands, health professionals in hospitals or residential care frequently lack the time and knowledge to support this vital area effectively [23,24].

Swallowing dysfunction and poor oral health were identified as independent risk factors for mortality in frail older people within a systematic review [25] and within elderly populations in residential care [26]. Elderly people suffering from OD are more likely to present poor oral health and hygiene, and a high prevalence of edentulism, periodontal disease, and caries [27]. Further studies have shown that intensified oral hygiene strategies can significantly reduce the incidence of aspiration pneumonia in frail elderly people [13,28,29]. The maintenance of optimal oral health care also demonstrated a reduced risk of aspiration pneumonia in patients after stroke [30], while a systematic review found that such care slowed the progression of respiratory diseases among high-risk elderly people living in nursing homes [31]. Furthermore, younger populations, such as children with special needs, may show an increased prevalence of OD and are also at a higher risk for oral health problems compared to their typically developing peers [32]. Still, very few studies have reported on the effects of improved oral health in similar young populations.

To date, no systematic review has been conducted to determine the effects of oral health care interventions in people with OD. However, the literature affirms that (1) OD may lead to aspiration; (2) OD may also lead to poor oral health; and (3) as poor oral health may lead to increased oral pathogens, patients with OD are at an increased risk of developing aspiration pneumonia. Moreover, no “best practice” has been established for oral health interventions in people with OD to reduce the risk of aspiration pneumonia. Therefore, the purpose of this study was to systematically review the literature on oral health care interventions in both children and adults with OD and its effects on swallowing-related outcomes, the incidence of aspiration pneumonia, and oral health status.

## 2. Materials and Methods

The methodology and reporting of this systematic review were guided by the updated Preferred Reporting Items for Systematic Reviews and Meta-Analyses (PRISMA) 2020 statement and checklists [32].

### 2.1. Eligibility Criteria

To be eligible for inclusion, articles had to meet the following criteria: (1) study participants were diagnosed with OD; (2) interventions aimed at improving oral health; (3) both pre- and post-intervention measurements were performed and reported on; (4) sample sizes of five or more participants, and (5) studies were published in English. Only original articles were included. Conference abstracts, reviews, case reports, student dissertations, and editorials were excluded. Study inclusion was not limited by study design. Studies focusing on dental caries and eating disorders (such as anorexia, bulimia, and/or behavioral eating aversions) were beyond the scope of this review.

### 2.2. Information Sources and Search Strategies

A literature search was conducted across four electronic databases: CINAHL, Embase, PsycINFO, and PubMed. All publication dates up to 19 July 2021 were included. Supplementary search strategies, such as cross-checking reference lists by hand, were also used to identify studies. Two categories of search terms were used in combination: terms related to (1) oral health, oral hygiene, mouth care, or dental care, and (2) swallowing, dysphagia, deglutition disorders, or feeding and synonyms. The electronic search strategies using subject headings (e.g., MeSH and thesaurus terms) are listed in Table 1 for each database.

### 2.3. (Data) Selection Process

Two independent raters reviewed all records and original articles from the literature searches for eligibility, utilizing the agreed-upon inclusion criteria. The inclusion of articles was based on consensus between the raters. An additional reviewer was consulted if agreement could not be reached between the first two reviewers. A methodological quality assessment was also rated by two independent researchers, after which 100% consensus was reached with the participation of a third reviewer in cases of disagreement.

### 2.4. Methodological Quality

The methodological quality of the included studies was assessed by the QualSyst critical appraisal tool [33]. The QualSyst tool consists of 14 questions with a three-point ordinal scoring system (yes = 2, partial = 1, and no = 0), providing a systematic, reproducible, and quantitative analysis of the research quality across a range of study designs from which a total score can be converted into a percentage score. As its standard, a QualSyst score >80% was interpreted as strong, 60–79% as good, 50–59% as adequate, and <50% as poor methodological quality [33]. Studies with poor methodological quality were excluded from further analysis. The level of evidence was classified in accordance with the National Health and Medical Research Council’s (NHMRC) evidence hierarchy levels [34].

### 2.5. Data, Items, Risk of Bias, and Synthesis of Results

The data collection process was supported by comprehensive data extraction forms to consistently withdraw information from all studies. The extracted data included: (1) intervention(s); (2) the study design (NHMRC level of evidence) and methodological quality (QualSyst score); (3) group descriptives (diagnostic group(s), age, gender, sample size); (4) the definitions used for oropharyngeal dysphagia and oral health; (5) the inclusion and exclusion criteria; (6) outcome measures related to swallowing, the incidence of (aspiration) pneumonia, and oral health; and (7) the main findings. The data were extrapolated and synthesized into different categories. No evident bias in the article selection or methodological study quality rating was present, as the reviewers did not have formal or informal affiliations with any of the included studies’ authors.

### 2.6. Synthesis Methods: Meta-Analyses

The general approach to data synthesis was to use a random-effects model to provide an average effect across all studies. Options for meta-analysis were considered only when studies of similar comparisons reported the same or comparable outcomes.

## 3. Results

### 3.1. Study Selection

A total of 4374 records were retrieved from four electronic databases: CINAHL (*n* = 645), Embase (*n* = 1923), PsycINFO (*n* = 44), and PubMed (*n* = 1762). Once duplicates were removed, the remaining 3681 titles and abstracts were assessed for eligibility using predefined inclusion and exclusion criteria. Following this, a total of 69 studies were assessed for eligibility using full texts; 17 studies were not original articles; 23 studies did not report on pre- and post-intervention outcomes (related to swallowing, the incidence of (aspiration) pneumonia, or oral health outcomes); 11 studies included participants without a confirmed diagnosis of OD; 9 studies did not report any kind of oral health intervention; and 1 study was not about oral health. Finally, eight original studies were included in this review. Figure 1 displays the flowchart of the selection process according to PRISMA 2020 [32].

### 3.2. Methodological Quality Assessment

The methodological quality of six studies was ranked as “strong” [35,36,37,38,39,40], and two studies were ranked as “good” [41,42]; no studies were excluded due to poor methodological study quality. Four studies were classified as level II evidence [35,36,37,41], and four were classified as level III evidence [38,39,40,42], based on the NHMRC evidence hierarchy level of evidence [34].

### 3.3. Study Characteristics and Results

A detailed overview of the included studies, reporting on study characteristics, study designs, methodological quality utilizing QualSyst, and information from the data extraction forms, is provided in Table 2. It was not possible to conduct a meta-analysis due to significant heterogeneity among the study designs (e.g., a randomized controlled trial versus a comparative study without a control group), subject populations (i.e., adults versus children), outcome measures (e.g., a clinical observation of oral health or the incidence of aspiration pneumonia), and interventions. Some studies also reported incomplete pre- and post-intervention outcome data.

### 3.4. Participants and Settings

The included studies reported on a total of 540 participants. The sample size per study ranged from 12 to 186 (mean 74 ± SD 58). The participants’ ages ranged from 3 to 101 years. Only one study included children (N = 24), representing 4% of all participants [41]. The intervention settings varied for adults, including in-patient hospital wards [38,42], rehabilitation centers [35,37,39], and nursing homes [36,40]. The one study that included children delivered the intervention in the child’s home [41]. Data were retrieved from studies conducted across seven countries and five continents.

### 3.5. Research Designs

All studies used a design in which two or more groups were compared. Four studies were classified as a controlled trials with randomization [35,36,37,41], two studies were comparative studies with concurrent controls and allocation not randomized [39,40], and two studies were comparative studies with historical controls [38,42]. Most studies (6/8) included a comparison group with patients with OD [35,36,37,38,41,42]. One study included people without OD as a comparison [39], and one study used two intervention groups: people with OD and people with impaired oral hygiene [40]. Six studies compared an enhanced or experimental oral health intervention with a standard oral health care intervention [35,36,38,39,41,42], and two studies compared three intervention methods [37,40]. The durations of the studies ranged from 1 week [39] to one year [36], with a median of 16 weeks.

### 3.6. Outcome Measures

Outcome measures varied greatly between studies. Swallowing-related outcomes on oral intake were assessed by the Functional Oral Intake Scale (FOIS) [35,36], patients’ nutritional status was described with the Mini-Nutritional Assessment-Short Form (MNA-SF) [35,38], and nasogastric tube removal was reported on [35,42]. Two studies had no outcomes related to swallowing [37,41]. The incidence of (aspiration) pneumonia was established by an instrumental assessment (thorax X-ray) [42] or by notes from medical records [36,39,41]. Two studies reported respiratory infections [38], pneumonia (not specified) [37], or cough frequency during mealtimes [40]. One study had no outcomes related to (aspiration) pneumonia [35]. Related to oral health, three studies performed objective oral health assessments by oral microfilm sampling for microbiological analyses [37], oral plaque scores [40], or calculus scores [41]; three studies used an observational measure to evaluate oral health: the Oral Health Assessment Tool (OHAT) [35,39] or the simplified Oral Hygiene Index (OHI-S) [38]; and two studies had no outcome measures on oral health [36,42]. Other outcome measures were hospital readmissions [38], survival rate [36], and mortality [38].

### 3.7. Interventions

The oral health care interventions for people with OD utilized in the studies were diverse but can be classified into three main groups: (1) oral disinfection by mouth rinse in combination with usual oral hygiene [36]; (2) intensified oral hygiene instruction or training [38,40], potentially combined with the free water protocol [39,42]; and (3) a combination of intensified oral health instruction and either topical oral disinfection [37] or tooth brushing with antibacterial toothpaste [35,41] and interdental cleaning [35].

### 3.8. Effects on Swallowing-Related Outcomes

Swallowing-related outcomes were assessed by nutritional status, oral intake measures, or the removal of a nasogastric tube in three studies with adult patients with OD [35,38,42]. One study reported a significant improvement in nutritional status by a minimal massive intervention (MMI) consisting of oral hygiene instruction plus texture-modified foods and caloric and protein supplementation assessed with MNA-SF [38]. One study reported non-significant but positive outcomes following an intensified oral health training intervention (combined with fluoride toothpaste and interdental brushing) in nutritional intake (assessed with FOIS) and an increased rate of nasogastric tube removal in the intervention group [35]. A similar trend was also identified in a study comparing regular oral health care and a free water protocol during a 40-day intervention period with a retrospective control group receiving inconsistent oral health care and placed on thickened liquids or liquid-restricted diets [42]. In that study, tube feeding was not required for the intervention group but was used in 18% of the retrospective control group participants [42].

### 3.9. Effects on Aspiration Pneumonia

Two studies reported reduced aspiration pneumonia incidence in adult patients with OD [36,42]. The study describing the effects of regular oral care combined with a free water protocol found statistically significant effects in favor of the intervention group after 40 days of treatment [42]. The second study reported a positive trend favoring the use of an oral disinfection mouth rinse (0.05% CHX) twice daily over regular oral health care without oral rinse after a one-year intervention [36]. Two studies found no incidence of (aspiration) pneumonia in either the intervention or control groups when using an intensified oral health program following intervention periods of one week [39] or six months [38]. Another study comparing three oral health interventions (oral hygiene instruction on the use of an electric toothbrush, additional mouth rinse, or nurse-assisted tooth brushing) [37] identified no significant group differences in hospital readmissions for pneumonia. The only study targeting a younger population identified a positive correlation between the presence of calculus and a history of aspiration pneumonia in children with gastrostomies [41].

### 3.10. Effect on Oral Health Status

Six studies reported outcomes on oral health or hygiene [35,37,38,39,40,41], of which three studies found statistically significant improvements [35,39,41]. The improvements were observed for an oral intervention of a dual-action whitening toothpaste [41], intensified oral hygiene instructions combined with fluoride toothpaste and interdental cleaning [35], or a nurse-led oral hygiene regime including twice-daily tooth brushing and a free water protocol [39]. The other studies found no significant between-group or pre-post differences in oral health using a 0.2% CHX oral rinse and assistance in addition to oral hygiene intervention [37], using oral hygiene recommendations (compared to standard clinical practice) [38], or with once-a-day tooth brushing (compared to upright feeding positioning or instruction in swallowing techniques) [40]. One study found that patients with OD presented poorer oral health outcomes pre- and post-intervention compared to the control group without OD and that independence for oral health care was associated with better oral health scores [39]. The measurements used varied from laboratory analysis [37,41] and OHAT/OHI [35,38,39] to a raw plaque score [40].

## 4. Discussion

The purpose of this systematic review was to synthesize the effects of oral health care interventions in adults and children with OD on swallowing-related outcomes, the incidence of aspiration pneumonia, and oral health status. Following the PRISMA 2020 guidelines for conducting systematic reviews [32], eight original studies were identified. The results from the methodological quality assessment of the studies utilizing the critical appraisal tool QualSyst [33] demonstrated that the evidence was at least of good study quality, with 75% of the studies assessed as “strong.” However, a meta-analysis was not feasible due to the heterogeneity of study designs, participant populations, outcome measures, interventions, and the incomplete reporting of study results. Current guidelines on meta-analyses indicate that if heterogeneity is not within reasonable limits the results of meta-analyses cannot be adequately interpreted, and, therefore, no meta-analysis should be conducted [44].

The effects of oral health interventions in people with OD have been reported on swallowing-related outcomes, aspiration pneumonia, and oral health status.

### 4.1. Swallowing-Related Outcomes

All studies reporting improvements in oral health interventions on swallowing-related outcomes [35,38,42] had intensified oral health hygiene in common. Only one study found significant improvements in nutritional status after six months of intensified oral hygiene instruction in combination with fluid thickening, texture-modified foods, and caloric and protein supplementation [38]. Due to the range of other strategies employed, the improvement in nutritional status was most likely not due to the oral hygiene intervention. The study using oral disinfection by mouth rinse in addition to usual oral hygiene care found a positive effect on the level of food intake, measured by the FOIS. This scale assesses a patient’s level of oral intake on a 7-point ordinal scale from 1 (nothing orally consumed) to 7 (normal diet). However, a progression in oral intake could not be expected due to the age of the participants [36]. Nasogastric tube removal was another positive effect related to swallowing outcomes but was not directly related to improved oral intake [35]. In general, to improve oral intake, alternative approaches to oral health may be of greater utility, such as OD intervention, dentition, food consistency modification, and/or caloric supplementation [45].

### 4.2. Aspiration Pneumonia

The incidence of aspiration pneumonia in adults with OD was reduced by using intensified oral hygiene instruction in combination with a free water protocol [42]. Good oral hygiene in the free water protocol is an important condition to ensure that the oral cavity is as free as possible from pathogens before water consumption. A previous study utilizing a free water protocol found that it did not result in an increased incidence of lung complications [46]. Another study employed topical oral disinfection (0.05% CHX) in combination with usual oral hygiene and found a positive trend in the reduction of aspiration pneumonia [36]. The positive results in these studies suggest that maintaining a clean mouth prevents the build-up of oral pathogens in people with OD [36,42]. This finding aligns with the positive correlation between good oral health and hygiene, reduced respiratory infections, and the occurrence of aspiration pneumonia, as described in the literature in elderly people in nursing homes and hospitals [9,11,20,25,29] and in the younger age group [41]. The benefits of CHX over water as a mouth rinse to prevent aspiration pneumonia in people with OD in nursing homes remains unclear. Factors such as compliance or the taste of CHX solutions could bias the results. Moreover, the concentrations of CHX varied between 0.05% [36] and 0.20% [37].

A further three studies did not have any events of (aspiration) pneumonia in either the intervention or control groups [37,38,39]. However, the current review provides limited evidence that regular oral health care and the free water protocol in addition to dysphagia intervention may reduce the incidence of aspiration pneumonia. Overall, the effects of ensuring a clean oral cavity after meals may be beneficial in preventing aspirations of residual food from the mouth.

### 4.3. Oral Health Status

Differences between the study outcome results may be due to the use of different treatments, outcome measures, and study durations. Three studies found significant improvements in oral health status using a combination of intensified oral hygiene instruction and tooth brushing with an antibacterial toothpaste or in combination with an additional water rinse or free water protocol [35,39,41]. Two other studies using intensified oral hygiene instruction [38] combined with oral disinfection [37] did not result in statistically significant improvements in oral health. The study outcomes and the short intervention programs with a limited number of treatment sessions may have resulted in the absence of these positive outcomes. Tooth brushing twice daily combined with oral rinsing is in line with the oral health recommendations for the general population [17] and a recent review [19]. As people with OD have demonstrably worse oral health compared to people without OD [25,30,39,41], giving intensified and comprehensive oral hygiene instructions to people with OD may be recommended, especially for those who are dependent on assistance in daily oral health care [25,29]. People with OD who require support from others showed worse oral health care compared with the general population [39]. Several studies were conducted within hospital or residential care settings where health professionals were required to assist with oral hygiene. Several factors may affect the efficacy of a caregiver’s intervention in oral hygiene: a lack of knowledge, skills, or time [47,48]. One study found a high feasibility for intensified oral hygiene and rinsing training in nursing home staff [40] alongside improved oral health outcomes for residents, suggesting that where increased training is provided, improved adherence and outcomes can be achieved. Increased awareness among health care professionals on the importance of oral health in patients with OD is required, as patients with OD suffer from worse oral health and a greater risk of aspiration pneumonia compared to controls [25,30,39,41].

### 4.4. Additional Factors in Oral Health Interventions

The duration of the studies varied considerably. In studies with an intervention of up to three weeks [37,39], aspiration pneumonia was not diagnosed in either the intervention or control groups, as the intervention period was too short for the occurrence of aspiration pneumonia. In contrast, an improvement in oral health was found in this brief period. On the other hand, an increased duration of oral health interventions did not necessarily result in improved outcomes for aspiration pneumonia and oral intake, as demonstrated by the study over a 12-month intervention period. The authors found only small between-group differences, likely due to a high participant drop-out rate and mortality (respectively, 44% and 29%) [36]. These high drop-out rates were explained by the study duration demanding a high level of motivation and commitment from the participants and intervention assistants, the reported bad taste of the 0.05% CHX rinse, and practical issues within the nursing homes (i.e., high staff turnover leading to shortages of trained staff and a need to prioritize competing demands). These additional external factors identified within the 12-month study should not be overlooked when determining the effects of oral health or hygiene interventions, particularly in relation to longer-term interventions.

Oral hygiene seems to be a simple and cost-effective method for people with OD, especially for those who are dependent on oral health care as provided by others and who are susceptible to aspiration pneumonia [20]. Although guidelines and educational courses on oral hygiene for caregivers and health professionals are essential, as has been previously highlighted [40], the benefits of only providing recommendations or oral hygiene instructions to people with OD without further assistance or training seems limited [35,38].

In line with previous findings that no international consensus exists for the definition of OD [49], a variety of criteria to define OD were found in the included articles. OD was mostly described in broad terms, (i.e., the presence of swallowing problems with or without a classification scale, cough during swallowing, or lacking any further descriptions), with only one study [36] providing a detailed definition of OD [43]. Furthermore, various instruments and outcome variables were used in the included studies to confirm OD, with only two studies using the instrumental examinations considered the ‘gold standard’ to confirm aspiration and dysphagia [50]: the videofluoroscopic swallow study (VFSS) [39] and the fiberoptic endoscopic evaluation of swallowing (FEES) [40]. This may have impacted the limited support found in this review for oral health interventions improving swallowing-related outcomes. Other studies [36,37,38] used a range of other tests and scales, resulting in different outcome measures and the potential to introduce bias [51]. The broad range of definitions of dysphagia, and diagnostic and treatment evaluation measures found in this review supports the need for increased consistency and consensus within the field of OD to improve further research and contribute to the best practice.

Although children with additional support needs, including those with OD, are at an increased risk of poor oral health care [5,32,52], there is limited information regarding effective oral health interventions for this population, with only one study [41] in this review targeting a child population. These children may experience increased challenges concerning oral health and dental care [22], and the lack of methodologically robust studies in this area represents a considerable barrier to ensuring optimal oral hygiene and reducing their risk of lung infections.

### 4.5. Limitations of Research and Recommendations for Future Research

This systematic review has some limitations, despite the rigorous reviewing process following PRISMA guidelines [32] and the measures taken to reduce bias. The final number of included evidence-based articles is limited, and many studies showed some methodological problems; some study designs were weakened by the lack of a control group receiving no therapy, and in most studies, the sample size was relatively small. Further, statistical pooling of the data was not possible for this review due to the heterogeneity of the study designs, outcome measures, and interventions. Only studies published in English were included in this review, whereas studies published in other languages could have added valuable findings. In addition, the authors used different terminology when describing lower respiratory infections (e.g., pneumonia, aspiration pneumonia, and lung infections) and may have provided insufficient details when defining aspiration pneumonia and reporting on the diagnostic tools to differentiate between distinct types of lung infections. Therefore, the conclusions from the included studies cannot be generalized easily or compared to one another because of the diversity in subject characteristics, assessment instruments, and interventions.

Further research utilizing randomized controlled trial study designs is recommended to fully evaluate the promising interventions for the reduction of aspiration pneumonia incidence and improving the oral hygiene of people with OD. This could then allow for findings to be summarized in line with the highest level of evidence by conducting meta-analyses. In particular, more studies are required to investigate the effects on oral health of CHX topical rinse in addition to tooth brushing, including the optimal concentration and duration of use. From such studies, an evidence-based oral hygiene care protocol could be developed to improve oral health to reduce the risk of developing aspiration pneumonia. Finally, as minimal data were available for younger participant populations, more studies should focus on oral health interventions in children with OD.

## 5. Conclusions

This review summarized the literature on the effects of oral health care interventions in adults and children with OD and its effects on swallowing-related outcomes, the incidence of aspiration pneumonia, and oral health status. Several oral health interventions provided limited evidence of reduced aspiration pneumonia incidence and/or improved oral health in people with OD. The most promising statistically significant intervention to prevent aspiration pneumonia in adults after brain injury in a hospital setting was twice daily oral cleaning in combination with the free water protocol. Although limited data were available for children, studies in adults indicated that oral cleaning, twice a day, with an antibacterial toothpaste in combination with interdental cleaning and the free water protocol proved to be the best intervention to improve oral health. The effect of improved oral health status on swallowing-related outcomes could not be established. Increasing the awareness and importance of oral health and the use of practical oral hygiene guidelines is recommended for both health professionals and the carers involved in the daily care of people with OD.

## Figures and Tables

**Figure 1 jcm-11-03521-f001:**
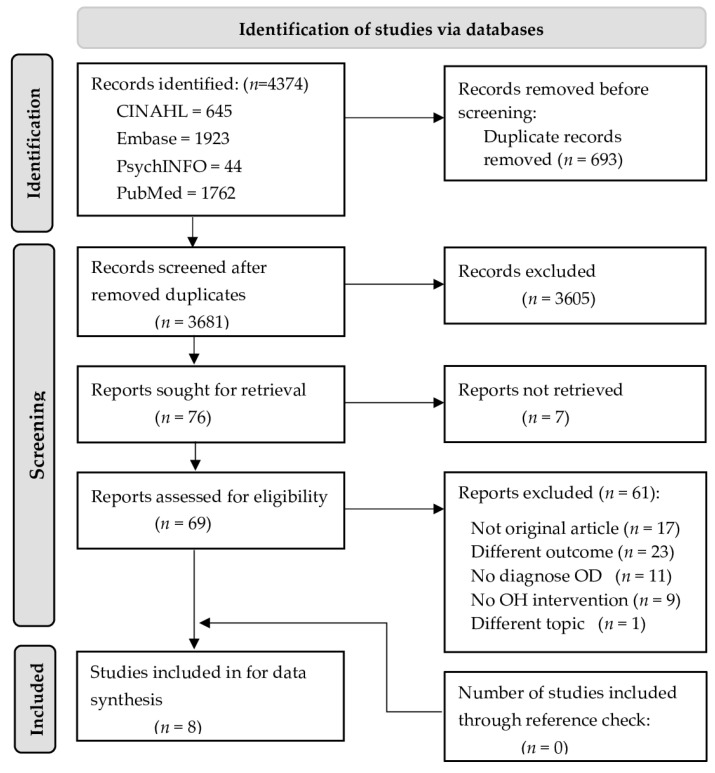
Flow diagram of the review process according to the Preferred Reporting Items for Systematic Reviews and Meta-Analyses (PRISMA) [32]. Legend: OD = oropharyngeal dysphagia; OH = oral health.

**Table 1 jcm-11-03521-t001:** Search strategies for each literature database.

Database and Search Terms	Number of Records
**Cinahl:** ((MH “Oral Health”) OR (MH “Oral Health (Iowa NOC)”) OR (MH “Oral Health (Omaha)”) OR (MH “Oral Health Maintenance (Iowa NIC)”) OR (MH “Oral Health Promotion (Iowa NIC)”) OR (MH “Oral Health Restoration (Iowa NIC)”) OR (MH “Oral Hygiene”) OR (MH “Self-Care: Oral Hygiene (Iowa NOC)”) OR (MH “Mouth Care”) OR (MH “Mouth Care (Saba CCC)”) OR (MH “Dental Care”) OR (MH “Dental Care for Aged”) OR (MH “Dental Care for Children”) OR (MH “Dental Care for Chronically Ill”) OR (MH “Dental Care for Disabled”)) AND ((MH “Deglutition”) OR (MH “Deglutition Disorders”) OR (MH “Feeding of Disabled”) OR (MH “Feeding and Eating Disorders of Childhood”) OR (MH “Infant Feeding”) OR (MH “Infant Feeding, Supplemental”) OR (MH “Parenteral Feeding (Saba CCC)”) OR (MH “Infant Feeding Pattern Impairment (Saba CCC)”) OR (MH “Ineffective Infant Feeding Pattern (NANDA)”) OR (MH “Feeding Self Care Deficit (NANDA)”) OR (MH “Feeding Methods”) OR (MH “Eating Behavior”) OR (MH “Eating Disorders Management (Iowa NIC)”) OR (MH “Eating Disorders”) OR (MH “Eating”))	645
**Embase:** (mouth hygiene/OR dental health/OR preventive dentistry/OR dental procedure/) AND (swallowing/OR dysphagia/OR feeding/OR feeding disorder/OR feeding difficulty/OR feeding behavior/OR eating/OR eating disorder/)	1923
**PsycINFO:** (Oral Health/OR Dental Health/) AND (swallowing/OR dysphagia/OR eating behavior/OR eating disorders/)	44
**PubMed:** (“Oral Health”[Mesh] OR “Oral Hygiene”[Mesh] OR “Dental Care”[Mesh] OR “Dental Care for Children”[Mesh] OR “Dental Care for Chronically Ill”[Mesh] OR “Dental Care for Aged”[Mesh] OR “Dental Care for Disabled”[Mesh] OR “Comprehensive Dental Care”[Mesh]) AND (“Deglutition”[Mesh] OR “Deglutition Disorders”[Mesh] OR “Feeding and Eating Disorders”[Mesh] OR “Feeding Behavior”[Mesh] OR “Feeding and Eating disorders of Childhood”[Mesh])	1762

**Table 2 jcm-11-03521-t002:** Study characteristics of the eight oral health interventions in oropharyngeal dysphagia.

Study	Intervention(s) Study Duration	Study Design ^a^ QualSyst Score ^b^	Group/Participants’ Descriptives (Age, Gender (Mean ± SD))	Oropharyngeal Dysphagia and Oral Health Definition(s) Inclusion/Exclusion Criteria	Outcome Measure(s)	Main Findings
Brown et al. (USA, 2006) [41]	To assess the anti-calculus benefit of Crest Dual Action Whitening Toothpaste compared to a control fluoride toothpaste, in children with gastrostomy. Instructed caregivers brushed subjects’ teeth twice-daily (for at least 45 s). Study duration: 2 × 8 wks.	NHMRC Level II QualSyst 75% (21/28)	Children with gastrostomy (GT) at home *Total group * N = 24; lost to follow-up: *n* = 3 M = 15; F = 7 Mean age: 7.2 ± 2.6 yrs. Cross-over 2 × 12 participants Intervention 1: Crest Dual Action Whitening Toothpaste; Intervention 2: Fluoride toothpaste.	OD as per gastrostomy -Dysphagia definition: problems of oral feeding and swallowing. -Oral health definition: effective oral hygiene. Professional calculus removal. -Inclusion: GT for ≥1 yr.; age 3–12 yrs.; enough erupted teeth for scoring purposes; daily oral hygiene by a caregiver; no professional dental prophylaxis within 3 mths. -Exclusion: allergy to components of study dentifrices; untreated oral conditions (e.g., caries).	-Oral health: Supragingival calculus using Volpe-Manhold Index (VMI) score -Incidence of aspiration pneumonia	Crest Dual Action Whitening Toothpaste reduced significantly supragingival calculus deposits by 58% compared to control fluoride toothpaste (*p* < 0.001). Calculus levels of the total group decreased by 68% over the study duration irrespective of dentifrice (*p*< 0.05). Calculus was significantly related to history of aspiration pneumonia (*p* ≤ 0.03). Lower baseline calculus scores were correlated with a greater number of tooth brushings per day (R^2^ = 0.47; *p* = 0.001).
Chen et al. (Taiwan, 2019) [35]	To evaluate the effect of oral health training (Bass method for tooth brushing, dental floss, interdental brushing, fluoride toothpaste) three times a wk. before swallowing therapy, additional to usual oral care. The control group received usual oral care (e.g., tooth brushing or sponge stick cleaning) twice-daily and an instructional manual to promote oral intake. Both groups received swallowing therapy. Study duration: 3 wks.	NHMRC Level II QualSyst 83% (20/24)	Patients with dysphagia after first-time stroke with a nasogastric tube in a rehabilitation centre. *Total group* N = 66 M = 43; F = 23 Mean age: N.R. G1 *Intervention:* Oral health training + swallowing training (*n* = 33) Age: ≥65 yrs. (*n* = 18) <65 yrs. (*n* = 15) G2 *Controls*: Usual oral care + instructional manual to promote oral intake (*n* = 33) Age: ≥65 yrs. (*n* = 18) <65 yrs. (*n* = 15)	OD as per not specified -Dysphagia definition: chewing and swallowing disorders. -Oral health definition: oral hygiene and a good oral state. -Inclusion: dysphagia following a first-time stroke; swallowing treatment; able to communicate in Chinese (Mandarin or Taiwanese); comply with instructions. -Exclusion: history of dysphagia due to oral cancer/head and neck cancer and/or ≥ 6 mths swallowing treatment.	-Oral health: OHAT -Oral intake: FOIS -Nutritional status: MNA-SF -Rate of nasogastric tube removal	Oral health training showed significant oral health improvements (OHAT) compared to usual care (*p* < 0.001). The intervention group had a higher, but non-significant FOIS score, for group difference (3.94 vs. 3.52; (*p* > 0.05), and for pre-posttreatment 3.15 vs. 3.94 There was no significant group and pre-post difference in nutritional status. Nasogastric tube removal was 21.2 % in the intervention group versus 6.1 % in the control group (not significant). The oral health program may improve oral health and maintain oral intake.
Hollaar et al. (Netherlands, 2017) [36]	To assess whether daily application of a 0.05% chlorhexidine (CHX) oral rinse solution, twice-daily in addition to usual oral hygiene, is effective in reducing the incidence of aspiration pneumonia. The control group received usual oral hygiene without the addition of an oral rinse. Patients were assisted by nurses if needed. Study duration: 1 yr.	NHMRC Level II QualSyst 88% (21/24)	Patients with dysphagia and physical disability in an in-patient nursing home. *Total group* N = 103 G1 *Intervention*: Usual oral hygiene + 0.05% CHX oral rinse (*n* = 52; lost to follow-up: *n* = 37) M = 25; F = 27 Mean age = 79.4 ± 8.9 yrs. G2 *Controls:* Usual oral hygiene (*n* = 51; lost to follow-up: *n* = 17) M = 26; F = 25 Mean age = 81.7 ± 9.03 yrs.	OD as per FOIS (level 1–6). -Dysphagia definition: difficulty with any stage of swallowing and dysfunction in any stage of oral intake; includes any difficulty in the passage of food, liquid, or medicine during any stage of swallowing that impairs the client’s ability to swallow independently or safely [43]. -Oral health definition: oral hygiene care, such as brushing teeth after each meal, cleansing dentures once daily, and professional oral healthcare once weekly. -Inclusion: age ≥65 yrs; physically disabled; diagnosed with dysphagia. -Exclusion: cognitively impaired; coma or vegetative state; terminally ill; dependent on mechanical ventilation; in daycare or short-term care; already using an oral hygiene care solution.	-Incidence of aspiration pneumonia -Survival rate Oral intake: FOIS	Daily use of 0.05% CHX oral rinse did not significantly reduce the incidence of aspiration pneumonia (*p* = 0.571), although a positive trend was found. High rate of dropouts in the intervention group (44% ) FOIS-level showed a significant risk of the incidence of aspiration pneumonia (*p* = 0.036).
Lam et al. (Hong Kong, 2013) [37]	To evaluate the effect of three oral hygiene interventions on opportunistic pathogens in patients after stroke. Patients were divided into three groups: (G1) oral hygiene instruction (OHI) and electric toothbrush only; (G2) OHI and 0.2% CHX mouth rinse twice-daily; (G3) OHI, 0.2% CHX mouth rinse twice-daily and nurse-assisted tooth brushing twice weekly. Study duration: 3 wks.	NHMRC Level II QualSyst 88% (23/26)	Patients with dysphagia after moderate to severe stroke (Barthel Index < 70) in a stroke rehabilitation centre. *Total group* N = 81; Age: >50 yrs. Gender: N.R. Mean age: N.R. G1: OHI (*n* = 25); G2: OHI + 0.2% CHX oral rinse (*n* = 26); G3: OHI + 0.2% CHX oral rinse + assisted tooth brushing (*n* = 30)	OD as per Royal Brisbane Hospital Outcome Measure for Swallowing. -Dysphagia definition: swallowing disability. -Oral health definition: good oral hygiene and professional oral health intervention. -Inclusion: moderate to severe stroke (Barthel Index <70); age >50 yrs.; admission to stroke rehabilitation ward ≤7 days earlier. -Exclusion: mild stroke; edentulism; communication difficulties; indwelling nasogastric tube.	-Oral health: prevalence of oral opportunistic pathogens by oral microbiological samples -Incidence of pneumonia	No significant intergroup differences were found in oral pathogens. Total counts of all opportunistic pathogens were significantly decreased in the OHI group (*p*= 0.032). No incidence of pneumonia was found. 0.2% CHX and assisted tooth brushing were found to have little effect on oral opportunistic pathogens during the in-hospital rehabilitation period.
Martín et al. (Spain, 2018) [38]	To assess the effect of a minimal massive intervention (MMI) in reducing nutritional and respiratory complications in elderly hospitalized patients with OD. MMI consisted of: a) fluid thickening and texture-modified foods; b) caloric and protein supplementation; and c) oral health and hygiene recommendations. The control group followed standard clinical practice without MMI. Study duration: 6 mths.	NHMRC Level III-3 QualSyst 91% (20/22)	Elderly with OD *Total group* N = 186 G1 *Intervention:* MMI (*n* = 62) M = 53%; F = 47% Mean age = 84.87 ± 6.02 yrs. G2 *Controls (retrospective)*: Standard clinical practice (*n* = 124) M = 53%; F = 47% Mean age = 84.42 ± 5.31 yrs.	OD as per V-VST. -Dysphagia definition: swallowing dysfunction that can include tracheobronchial aspirations. OD is related to impaired safety of swallow, or the incapability to protect the respiratory airway effectively. Geriatric syndrome -Oral health definition: oral hygiene, tooth brushing frequency, use of mouthwashes, use of dentures, and dentist visit. -Inclusion: age ≥70 yrs. -Exclusion: severe dementia (Global Deterioration Scale ≥6); discharged from intensive care unit; severe functional dependence (Barthel Index ≤40); low survival probability (Walter score ≥6).	-Hospital readmissions -Respiratory infections -Mortality -Nutritional status: MNA-SF -Oral Health: OHI-S -Functionality: Barthel index	Significant group differences in favor of the intervention group: -decreased hospital readmission (*p* = 0.001); -higher survival rate (84.13% vs. 70.96%). No significant group differences for readmissions for pneumonia. Within the intervention group: -Improved functional capacity (*p* = 0.007); -Improved nutritional status (*p* = 0.0038); -No improved oral health (*p* = 0.095).
Murray & Scholten. (Australia, 2018) [39]	To determine whether a simple oral hygiene protocol improves the oral health. A nurse-led oral hygiene regime included twice-daily tooth brushing and mouth rinsing after lunch. OD G1 had additionally no water restriction OD G2 received only thick fluids. The control group without OD received regular fluids. Study duration: 1 wk.	NHMRC Level III-3 QualSyst 91% (20/22)	Patients with OD after stroke in rehabilitation setting. *Total sample* N = 12 M = 9; F = 3 Mean age = 79 ±6.9 yrs. *Interventions* G1 Oral hygiene regime + free water protocol *n* = 7. G2 Oral hygiene regime + thickened liquids only *n* = 5. Gender and age: N.R. *Controls (no OD)* Oral hygiene regime *n* = 77 M = 48; F = 29; Mean age: 69 ±11.3 yrs.	OD as per VFSS of fluid, 150 mL water test, mealtime observation. -Dysphagia definition: facial paresis, tongue weakness, and poor oral sensation resulting in poor control of dentures, altered chewing, and reduced clearance of food from the oral cavity. -Oral health definition: oral hygiene and health of lips, tongue, and oral mucosa. -Inclusion total group: stroke; medical stability; full oral diet. For OD group: aspiration of thin liquid, but safe consumption of at least pureed food and one consistency of thickened liquids. -Exclusion total group: progressive neurological disease; acute illness; requiring fluid supplementation or fluid restriction. For OD group Chronic Obstructive Pulmonary Disease (COPD); immunosuppression.	-Oral Health: OHAT -Incidence of aspiration pneumonia	Oral health improved significantly (59%) in the intervention group compared to the control group. No patients developed aspiration pneumonia. Patients with OD had worse oral health compared to controls (no OD) pre- and post-intervention *p* = 0.027 vs. *p* = 0.023. Patients with OD improved on oral health (*p* = 0.024) compared to the controls (*p* = 0.282). Independence for oral care was associated with better oral health scores (*p* = 0.027).
Quagliarello et al. (USA, 2009) [40]	To test intervention protocols for feasibility, staff adherence, and effectiveness in reducing pneumonia risk factors (impaired oral hygiene and swallowing difficulty) in nursing home residents. Intervention group OH: (G1) manual oral brushing morning + 0.12% CHX rinse evening; (G2) manual oral brushing morning + 0.12% CHX rinse morning/evening; (G3) manual oral brushing morning/evening + 0.12% CHX rinse every morning/evening. Intervention group OD: (G4); feeding position > 90° with each meal; (G5) Instruction in swallowing techniques with each meal; (G6) Manual oral brushing every morning. Study duration: 3 mths.	NHMRC Level III-2 QualSyst 83% (20/24)	People with swallowing difficulties and impaired oral hygiene in nursing home residents. *Total sample* N = 52 (M = 10%; F = 90%) Mean age = 86.0 ± 7.8 yrs. *Group OH* (*n* = 30) Inclusion: impaired oral hygiene Age and gender: N.R. G1: manual oral brushing morning + 0.12% CHX rinse evening (*n* = 10); G2: manual oral brushing morning + 0.12% CHX rinse morning/evening (*n* = 10); G3: manual oral brushing morning/evening + 0.12% CHX rinse every morning/evening (*n* = 10). *Group OD* (*n* = 22) Inclusion: OD Age and Gender: N.R. G4: Upright feeding positioning: *n* = 7. G5: Manual oral brushing: *n* = 8 G6: Instruction in swallowing techniques: *n* = 7	OD as per FEES: retention of a 5-mL bolus in the vallecula or piriform sinus (mild impairment), laryngeal penetration of the bolus in the laryngeal vestibule but above the vocal folds (moderate impairment), or aspiration of the bolus below the level of the vocal folds (severe impairment). -Dysphagia definition: swallowing difficulty according to the FEES criteria. -Oral health definition: oral hygiene; low plaque score -Inclusion: age >65 yrs., plaque score >1.0, cough during swallowing during at least one meal in a week. -Exclusion: residents <4 wks.; short-term rehabilitation; estimated survival ≤6 mths (by nursing staff); tube-fed; tracheostomy.	-Feasibility: time to complete the protocol -Staff Adherence: high, moderate, or low -Cough frequency during swallowing during at least one meal within the previous week -Oral health: plaque control on a 4 point ordinal scale of six teeth	High feasibility for all interventions, except for instruction in swallowing techniques (47.6%). High staff adherence was achieved in all interventions, except Instruction in swallowing techniques (73.1%). All OH interventions demonstrated high feasibility, high staff adherence, Group OH; Pre-post improvement of plaque score (*p* = 0.001); the combined brushing plus 0.12% CHX rinse twice-daily showed the highest plaque score reduction of 1.69. Group OD: Reduced episodes of cough were observed during swallowing in all groups: G1 (43%); G2 (75%); and G3 (43%). No intervention was significantly more effective than any of the other two interventions (*p* = 0.31). Daily manual oral brushing and upright feeding positioning demonstrated high feasibility, high staff adherence, and effectiveness in improving swallowing.
Seedat & Penn (South Africa, 2016) [42]	To investigate whether it was possible to reduce the occurrence of aspiration pneumonia for patients presenting with OD by implementing a regular routine of oral care. The intervention group received regular oral care and was not restricted from drinking water for half an hour after oral intake, but restricted for all other liquids. The control group received inconsistent oral care and were restricted to thickened liquids or liquid restricted diets. Both groups received dysphagia intervention. Study duration: 40 days.	NHMRC Level III-3 QualSyst 73% (16/22)	Patients after stroke or traumatic brain injury in government hospitals. *Total sample* N = 46 (Stroke *n* = 32 Brain injury *n* = 14) M = 50%; F= 50% Age: N.R. G1 *Intervention*: Regular oral care + free water protocol (*n* = 23). G2: *Controls (retrospective)*: Inconsistent oral care + restricted thickened liquids/liquid restricted diets (*n* = 23). Groups were matched for medical diagnoses No differences in gender between groups.	OD as per not specified. Dysphagia definition: difficulty swallowing food or drinking liquids. -Oral health definition: oral care and hygiene to reduce complications from both a dental and respiratory perspective. -Inclusion: stroke or traumatic brain injury (primary diagnosis). -Exclusion: aspiration pneumonia at start of the study.	-Aspiration pneumonia -Nasogastric tube	Regular oral care and free water provision combined with dysphagia intervention prevent aspiration pneumonia in patients with OD. A moderate association was established between aspiration pneumonia and group: (*p* = 0.0092) 30% of the controls presented aspiration pneumonia whereas none in the intervention group. Four persons in the control group got a nasogastric tube and none in the intervention group.

Legend: CHX: chlorhexidine; FEES: fiberoptic endoscopic evaluation of swallowing; FOIS: Functional Oral Intake Scale; MNA-SF: Mini-Nutritional Assessment-Short Form; N.R: Not reported; OD: oropharyngeal dysphagia; OH; oral health; OHAT: oral health assessment tool; OHI-s: Oral Health Index-Simplified; VFSS: videofluoroscopy; V-VST: Volume-Viscosity Swallow Test. ^a^ NHRMC hierarchy [34]: Level I Systematic reviews; Level II Randomized control trials; Level III–1 Pseudo-randomized control trials; Level III–2 Comparative studies with concurrent controls and allocation not randomized (cohort studies), case-control studies, or interrupted time series with a control group; Level III–3 Comparative studies with historical control, two or more single-arm studies, or interrupted time series without a control group; Level IV Case series. ^b^ Methodological quality (QualSyst) [33]: strong >80%; good 60–79%; adequate 50–59%; poor <50%.

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
