# Peer review of "Effects of Oral Health Interventions in People with Oropharyngeal Dysphagia: A Systematic Review"

_jcm, 2022, doi:10.3390/jcm11123521_

Round 1
Reviewer 1 Report
Thank you for inviting me to review this manuscript which presented a systematic review of oral healthcare interventions for individuals with oropharyngeal dysphagia. Oral healthcare is a critical and topical issue presently, receiving increasing attention in health and government. It seems so fundamental but in reality is so complex to get right. Continuing to publish good quality research in this area is paramount. I commend the authors on their endeavours.
This manuscript is is a well-designed and rigorous review with excellent descriptions of the methods. Several systematic reviews have previously been published about this topic and its relationship to aspiration pneumonia, so I believe the authors need to establish the gap and the uniqueness of their review more strongly. The inclusion of children (and dearth of studies in this field) and the focus on people with oropharyngeal dysphagia could be more strongly stated.
Some restructuring of the results and discussion sections would also increase readability and emphasise key findings better.
Please find below some more specific feedback.
Introduction:
Line 47 The relationship between OD, aspiration pneumonia and oral bacteria needs further research and a more sophisticated explanation - the oral microbiome is an intricate balance between good and bad bacteria, so it is not just the growth in bacteria that causes the risk - it is the imbalance between good and bad......Oropharyngeal dysphagia impacts this balance in many ways not just related to salivary flow.
Lines 76-89 could be synthesised and described more succinctly than is presented in this paragraph.
Lines 74, 75 The sentence about children with special needs seems out of place in this paragraph. Is there more literature that could be sourced to support the importance of oral care for children with OD? If not, it is important, given this is a unique contribution of this systematic review, to highlight the lack of studies. Could it have its own paragraph?
Lines 95,96 Given the systematic reviews that have been conducted in the elderly, frail, and stroke populations outlined in the previous paragraph, the gap and therefore the reason for this systematic review is not articulated strongly enough. The argument about it being across the lifespan and looking at outcomes not previously researched needs to be strengthened by highlighting this as the gap.
Methods:
Well explained
Results:
Line 185. In the description of the participants, please include a summary of the settings for the studies i.e. hospitalised, residential care, community
Line 228 The interventions need to be better described for the Murray et al and Seedat et al papers in that their interventions included introduced a free water protocol with the oral hygiene. A free water protocol is not just unrestricted access to water, but rather has strict rules to minimise aspiration pneumonia risk, so this needs to be responsibly described here and in the conclusion drawn in line 336 of the discussion.
Table 2 seems to have different reference numbers attributed to the papers than those that are cited in-text and in the reference list. Please review.
Discussion - sections 4.1-4.3 Start with a further reporting of detailed results. These sentences need to be removed from the discussion and summarised and interpreted more on a meta-level. For example, lines 290-296 could be synthesised with the results in 3.8. Then the discussion highlights that the oral health intervention itself may not have been the factor contributing to outcomes. Please consider this for sections 4.2 and 4.3 as well.
Line 377 begins a discussion of Other Factors so perhaps needs its own sub-heading rather than continuing on from Oral Health Status.
Author Response
Point-to-point response to Reviewer 1
We thank the reviewer for the constructive feedback. To improve readability and language, the manuscript has been checked by a native English speaker.
The numbers of the given lines refer to the manuscript in the format of the journal
- Several systematic reviews have previously been published about this topic and its relationship to aspiration pneumonia, so I believe the authors need to establish the gap and the uniqueness of their review more strongly.
We clarified this in the final paragraph of the Introduction. We refer to some reviews and add ‘review’ in the text.
See line 87 ‘To date, no systematic review has been conducted to determine the effects of oral health care interventions in people with OD.
- The inclusion of children (and dearth of studies in this field) and the focus on people with oropharyngeal dysphagia could be more strongly stated.
We clarified the topic of our review in the final paragraph of the Introduction and added a final comment under subheading 4.5.
See lines 81-85: ‘Furthermore, younger populations, such as children with special needs, may show an increased prevalence of OD and are also at higher risk for oral health problems compared to their typically developing peers [32]. Still, very few studies have reported on the effects of improved oral health in similar young populations.’
See lines 450-451: ‘Finally, as minimal data were available for younger participant populations, more studies should focus on oral health interventions in children with OD.’
Introduction:
- Line 47 The relationship between OD, aspiration pneumonia and oral bacteria needs further research and a more sophisticated explanation.
We improved the description of the relationship between OD, aspiration pneumonia, and oral health and have added new literature.
See lines 46-54
- Lines 76-89 could be synthesised and described more succinctly than is presented in this paragraph.
We revised and shortened the paragraph.
See lines 76-85
- Lines 74, 75 The sentence about children with special needs seems out of place in this paragraph. Is there more literature that could be sourced to support the importance of oral care for children with OD? If not, it is important, given this is a unique contribution of this systematic review, to highlight the lack of studies. Could it have its own paragraph?
We revised the paragraph to clarify research on younger populations.
See lines 81-85 and 450-451 and the response above.
- Lines 95,96 Given the systematic reviews that have been conducted in the elderly, frail, and stroke populations outlined in the previous paragraph, the gap and therefore the reason for this systematic review is not articulated strongly enough. The argument about it being across the lifespan and looking at outcomes not previously researched needs to be strengthened by highlighting this as the gap.
We highlighted that up to date, no systematic review has been conducted to determine the effects of oral health interventions in people with OD.
See line 87: ‘To date, no systematic review has been conducted to determine the effects of oral health care interventions in people with OD.
Results
- Line 185. In the description of the participants, please include a summary of the settings for the studies i.e. hospitalised, residential care, community
All settings have been added in the description of the participants (See 3.4; Participants and settings) and in Table 2.
- Line 228 The interventions need to be better described for the Murray et al and Seedat et al papers in that their interventions included introduced a free water protocol with the oral hygiene. A free water protocol is not just unrestricted access to water, but rather has strict rules to minimise aspiration pneumonia risk, so this needs to be responsibly described here and in the conclusion drawn in line 336 of the discussion.
We revised the description of the interventions of the free water protocol by Murray et al. and Seedat et al. throughout the manuscript (See: Table 2, paragraph 3.8 and the Discussion Section.
- Some restructuring of the results and discussion sections would also increase readability and emphasise key findings better.
We revised both the Results and Discussion sections to increase readability and emphasize our main conclusions. Moreover, the manuscript has been checked by a native English speaker.
- Table 2 seems to have different reference numbers attributed to the papers than those that are cited in-text and in the reference list. Please review.
We have corrected the reference numbers in Table 2 to be in line with the reference list.
Discussion
- sections 4.1-4.3 Start with a further reporting of detailed results. These sentences need to be removed from the discussion and summarised and interpreted more on a meta-level. For example, lines 290-296 could be synthesised with the results in 3.8.
Then the discussion highlights that the oral health intervention itself may not have been the factor contributing to outcomes. Please consider this for sections 4.2 and 4.3 as well.
We revised the Discussion Section in line with the reviewer’s suggestions. We also moved information from parts 4.1, 4.2, and 4.3 to 3.8, 3.9, and 3.10 respectively
Line 377 begins a discussion of Other Factors so perhaps needs its own sub-heading rather than continuing on from Oral Health Status.
We agreed with the reviewer to import another heading 4.4 (Additional factors in oral health interventions). This paragraph concerned the duration of the intervention and training of health care providers.

Reviewer 2 Report
Currently, a variety of oral health cares are being implemented for people with oropharyngeal dysphagia. However, as the authors state, ‘best practice’ has not been established and are based on rules of thumb. In this context, this paper is an excellent achievement in providing a benchmark for oral health care in the form of a systematic review. On the other hand, I believe there is room for this paper to be of higher quality. Please consider the following arguments from me.
- It is stated in the text that there is a 'significant heterogeneity', but please provide the results of an actual test of heterogeneity among the eight studies.
- Is it possible that the way of dealing with oropharyngeal dysphagia differs between developing children and older adults with functional decline due to aging? The author is correct that oropharyngeal dysphagia can happen to anyone, but in this final selecting papers where the subject age is known, most of the subjects are older adults. Is it reasonable to generalize these results to all ages?
- As the authors state, it is difficult to implement proper oral health care without assistance and training. Wouldn't it be an important perspective to know which individuals in each of the eight studies implemented oral health care?
Author Response
Point-to-point response to Reviewer 2
We thank the reviewer for the constructive feedback. To improve readability and language, the manuscript has been checked by a native English speaker.
-It is stated in the text that there is a 'significant heterogeneity', but please provide the results of an actual test of heterogeneity among the eight studies.
When selecting outcome measures for inclusion in meta-analysis, reducing heterogeneity between studies is a priority. If heterogeneity is not within reasonable limits, the results of meta-analyses cannot be adequately interpreted. Therefore, due to the heterogeneity in study designs (e.g., no control group), differences in outcome measures (e.g., clinical observation of oral health, incidence of aspiration pneumonia), differences in age groups (children versus adults), and insufficient data for meta-analyses (incomplete reporting on pre- and post-treatment outcome measurements), we did not perform any meta meta-analyses. Therefore, we also did not conduct a test of heterogeneity as in line with current guidelines on conducting meta-analyses, outcomes that are too disparate should not be combined in meta-analyses as unequivocal differences in effects may be obscured. Conducting a test of heterogeneity is useful when performing meta-analyses.
We clarified the rational for not conducting meta-analysis in paragraph 3.3 and first paragraph of the Discussion section.
Is it possible that the way of dealing with OD differs between developing children and older adults with functional decline due to aging? The author is correct that oropharyngeal dysphagia can happen to anyone, but in this final selecting papers where the subject age is known, most of the subjects are older adults. Is it reasonable to generalize these results to all ages?
Our review determined the effects of oral health interventions in people with OD across the lifespan, but as the reviewer already commented, only a single study in children met our inclusion criteria. We agree with the reviewer that not all recommendations may be generalized to children. Therefore, we added a comment in the Result section (para 3.9), Discussion section (para 4.5), and the Conclusion referring to the limited data available for younger populations.
As the authors state, it is difficult to implement proper oral healthcare without assistance and training. Wouldn't it be an important perspective to know which individuals in each of the eight studies implemented oral health care?
We added information, if available, about whom was involved in implementing oral health care in Table 2.

Reviewer 3 Report
This systematic review thoroughly investigates the impact oral hygiene has on OD, a disorder that affects a large portion of the human population. The abstract is clear and concise and the introduction sets up the paper nicely to show the vast patient population that are affected by OD and aspiration pneumonia. The introduction emphasizes why this paper is necessary.
I only have minor comments/edits
In the methods section 2.4: is the numbering system representing quantity? Or is there two number (2) by accident?
Results section 3.4” please change the SD to 59 so it stays with the same format of 2 numbers.
In table 2: please put column headers on each page. The headers are currently only on the first page of this table. Since the table is too large to fit on one page, the headers need to continue throughout the pages of this table.
Author Response
Point-to-point response to Reviewer 3
We thank the reviewer for the constructive feedback. To improve readability and language, the manuscript has been checked by a native English speaker. The numbers of the given lines refer to the manuscript in the format of the journal
In the methods section 2.4: is the numbering system representing quantity? Or is there two number (2) by accident?
We added some information about the QualSyst in paragraph 2.4.
See lines 143-145. The QualSyst tool consists of 14 questions with a three-point ordinal scoring system (yes = 2, partial = 1, and no = 0), providing a systematic, reproducible, and quantitative analysis of the research quality across a range of study designs, from which a total score can be converted into a percentage score.
Results section 3.4; please change the SD to 59 so it stays with the same format of 2 numbers.
We changed the SD. See line 196.
In table 2: please put column headers on each page. The headers are currently only on the first page of this table. Since the table is too large to fit on one page, the headers need to continue throughout the pages of this table.
We added headers on every page of the table.
